

# Sea Surface Salinity Short Term Variability in the Tropics

Frederick M. Bingham[1], Susannah Brodnitz[1]

[1]Center for Marine Science, University of North Carolina Wilmington, Wilmington, 28403-5928, USA

*Correspondence to*: Frederick M. Bingham (binghamf@uncw.edu)

**Abstract.** Using data from the Global Tropical Moored Buoy Array we study the validation process for satellite measurement of sea surface salinity (SSS). We compute short-term variability (STV) of SSS, variability on time scales of 5-14 days. It is

meant to be a proxy for subfootprint variability as seen by a satellite measuring SSS. We also compute representation error, which is meant to mimic the SSS satellite validation process where footprint averages are compared to pointwise in situ values. We present maps of these quantities over the tropical array. We also look at seasonality in the variability of SSS and find which months have maximum and minimum amounts. STV is driven at least partly by rainfall. Moorings exhibit larger STV during rainy periods than non-rainy ones. The same computations are also done using output from a high-resolution global ocean

model to see how it might be used to study the validation process. The model gives good estimates of STV, in line with the moorings, though tending to have smaller values.

**Plain text summary**. Satellite measurements of sea surface salinity (SSS) are compared with measurements in the ocean to verify the quality of the satellite data. SSS satellites measure average values over a footprint with size ~100 km, whereas ocean

values are usually taken at a single point in space and time. Using SSS data from a network of buoys across the global tropics we estimate the size of the mismatch between satellite and in place measurements to better understand the error structure of the satellite.

## 1 Introduction

Sea surface salinity (SSS) has been measured by satellite for more than a decade. Along the way there have been remarkable advances in the quality of the data and their applications (Reul et al., 2020; Vinogradova et al., 2019). SSS is measured by satellite using L-band radiometers, combined with ancillary measurement of SST, sea ice, rain rate and wind speed, and





corrections for such factors as galactic radiation, Faraday rotation in the atmosphere and radio frequency interference (Meissner et al., 2018; Olmedo et al., 2020).


As the database of satellite-based SSS measurements grows, the need to fully document the errors in the measurements gets more acute. Many of the sources of error are well known and quantified (Lagerloef et al., 2008; Meissner et al., 2018). However, an important source has not been as well studied, that of representation error (RE). The accuracy of SSS measurements is often assessed by comparison with individual in situ readings, such as might be taken by an Argo float (Abe

& Ebuchi, 2014; Kao et al., 2018b; Olmedo et al., 2017; Dinnat et al., 2019; etc.), mooring, glider or ship, a process known as validation. RE is when two measurements being compared do not represent the same quantity. That is, the validation measurement and the satellite measurement are mismatched somehow in scale or timing. In the case of comparisons with float measurements, there may be differences between satellite and validation measurement due not to error or inaccuracy in in situ instruments or to retrieval, but to a mismatch in scale between the two systems. One example of RE is that of subfootprint

variability (SFV; Boutin et al., 2016; Bingham, 2019). SFV occurs because the SSS satellite measurement is made over a large footprint, whereas individual float measurements are made at a single point. The footprint of the satellite is ~100 km in the case of the Aquarius satellite (Lagerloef et al., 2008) and ~40 km in the case of the Soil Moisture Active Passive (SMAP) satellite (Meissner et al., 2019). The other major SSS satellite, SMOS (Soil Moisture and Ocean Salinity) does not have a simple footprint due to its interferometric method of sensing and wide field of view.


SFV has been discussed in detail by Bingham (2019). That paper quantified SFV for a 100 km satellite footprint at two locations, the SPURS-1 (Salinity processes in the Upper ocean Regional Studies -1) region in the subtropical North Atlantic and the SPURS-2 region in the eastern tropical North Pacific, using a combination of drifter salinity, thermosalinograph and waveglider data. The paper computed not just SFV, but its impact on satellite SSS error at those two locations. The analysis

was further extended to include a variable footprint size by Bingham & Li (2020). One clear result of these two efforts is the difference between the two regions, and the time variability of SFV. The SPURS-2 region has a higher amount of SFV than SPURS-1, less dependence on footprint size, and less seasonal variability. The analysis of Bingham (2019) also included a comparison of SFV computed from in situ data with short-term variability (STV) computed from moorings located at the two sites. The two were similar in magnitude and had similar seasonality, indicating that STV from a mooring could be used as a

reasonable proxy for SFV from distributed in situ data. In this paper we take that conclusion and go further with it. We make use of data from the global tropical moored buoy array (GTMBA) to compute STV as a proxy for SFV over the global tropics and quantify SSS SFV, RE, and their magnitude, variability and geographic distribution.

Another type of RE that is commonly thought of is temporal aliasing. SSS satellites have a limited footprint extent and limited

temporal coverage. In the case of Aquarius, the satellite repeated every 7 days, whereas with SMAP it is 2-3 days (Reul et al., 2020). Thus, an in situ measurement may not be simultaneous with a satellite overpass in time, leading to a potential difference

between the two, or temporal aliasing. In this paper, due to our use of temporal sampling as a proxy for spatial, we cannot distinguish between SFV, or spatial aliasing, and temporal aliasing. We will briefly discuss the temporal aliasing issue though.

SSS data from the GTMBA have been used in the past for comparison with satellites. Most comparisons have been done at level 3 (L3; Bao et al., 2019; Tang et al., 2017; Qin et al., 2020; Tang et al., 2014), and some at L2 (Abe & Ebuchi, 2014; Kao et al., 2018a,b; Tang et al., 2014). For example, Bao et al. (2019) computed RMS differences and bias between mooring, satellite (SMOS and SMAP) and in situ gridded (EN4; Good et al., 2013) data, where the mooring data used were 8-day moving averages. Tang et al. (2017), computed similar statistical comparisons between moorings and SMAP, again using 8-

day average values. Qin et al. (2020) reported RMS error and bias between satellite SSS and a small set of moorings. While the GTMBA moorings have been a useful point of comparison for validation, as indicated by the number of studies we have just cited, they have not to date been used to study the process by which validation is carried out. As the mooring data are generally high quality, sampled at a high frequency, dispersed broadly over a diverse set of tropical regimes, and are placed very near the surface, they make an ideal platform for this.


One complementary aspect that we will study here is the use of high-resolution model output for exploring STV. Bingham (2019) and Bingham & Li (2020) both compared SFV from a high-resolution model (different from the one we will use here) and from in situ data and found that the two agreed reasonably well, especially in the subtropical SPURS-1 region. We would like to use such model output to study SFV on a global scale. The comparison of statistics from the moorings and the global

model can give us confidence that model output may be used for this purpose.

## 2 Data & Methods

We use two sources of in situ data. Velocity data from the OSCAR (Ocean Surface Current Analysis Realtime) dataset, and SSS from the GTMBA. We will also use data from the MITgcm (Massachusetts Institute of Technology general circulation

model) as described below.

In this paper we use practical salinity from the 1978 practical salinity scale (UNESCO, 1981). This scale is unitless, so, following Millero (1993), we do not use the terms "psu" or "pss" as a substitute for units.

**2.1 GTMBA SSS**





The GTMBA is a vast network of buoys stretching across the global tropics (Figure 1). It was originally set up in the mid-1980's to measure El Niño-related variability of the tropical Pacific (McPhaden et al., 1998; McPhaden et al., 2010) and has since been expanded to the Atlantic (Foltz et al., 2009) and Indian (McPhaden et al., 2009) basins. Most of the moorings have

sensors that measure SSS at ~1 m depth with a Sea-Bird SBE-37 microcat instrument (Freitag et al., 2018). GTMBA SSS measurements are reported hourly. No quality control was carried out beyond that done by the agencies operating the array. Bao et al., (2019) identified a small number of moorings with suspicious drifts in their SSS records. We examined the same records and did not judge them to be problematic. As the analysis done here uses short bursts of data to study variability, usually on time scales of ~7 days or less, absolute accuracy of the sensor is not crucial. SSS in the tropics tends to be quite

spiky, with many low outliers, (e.g. Bingham et al. (2021), their Figure 2) Overly stringent quality control could eliminate many valid data points and alter the statistics of the record.

Many of the GTMBA moorings also recorded precipitation using an R.M. Young: 50203 self-siphoning rain gauge (Freitag et al., 2018). The available records are at one-minute intervals, from which we computed hourly averages.

**2.2 OSCAR currents**

Ideally, in order to determine SFV, we would have a spatially distributed set of ocean measurements taken simultaneously like those from SPURS-1 and 2 (Bingham, 2019; Bingham & Li, 2020). Instead, we have intensively sampled time series of SSS measurements (Section 2.1) at a set of discrete locations. Our assumption is that one can substitute for the other. To tie space

and time together, we use the OSCAR dataset, which is an estimate of surface current derived from satellite altimetry, sea surface temperature and surface vector winds (Bonjean & Lagerloef, 2002). The values of surface current come on a 1/3° grid at 5-day intervals (ESR, 2009). We computed average speed (not average velocity) over the 1992-2020 time period at each mooring location (Figure 1). This average speed was then turned into a short-term ensemble time period by dividing100 km by the average speed. The time periods varied from 2.3 to 17 days, with a median value of 5 days. The main assumption is that

within the short-term ensemble time period, the mooring samples about a 100 km area of ocean at the given average speed, and that this 100 km sample gives an estimate of the SFV.

**2.3 MITgcm**

We will use the MITgcm with a latitude-longitude polar cap grid, the "LLC-4320" (Su et al., 2018). The nominal horizontal resolution is 2.3 km near the equator. The model output was available for the 1 November 2011 to 31 October 2012 time period. The model is free-running, i.e. no ocean data assimilation, and forced with 6-hour atmospheric fields from the ECMWF (European Centre for Medium-range Weather Forecasting) 0.14° atmospheric operational model analysis. We obtained the





SSS field from the model and extracted time series for each of the locations of the GTMBA moorings. We carried out many
of the same analyses with model SSS data as we did with the real mooring data – see below – with the exception of computing
the seasonal cycle. Only one year of output is not enough to get a robust estimate of the seasonal variability.

In addition, at all the mooring sites, we computed once daily values of SFV from the model grid surrounding each site. SFV
is computed as a gaussian weighted standard deviation using a 50 km decay scale, i.e. a 100 km footprint. The method is
similar to that of Bingham (2019), but using instantaneous regions of model grid values instead of 7-day in situ data.

## 2.3 Short-term variability

To compute STV from the SSS data, each record was divided into weekly evaluation times. At each of these times, we isolated
an ensemble of SSS measurements surrounding it in the time interval given by the ensemble time period computed from the
OSCAR data. The STV was computed as the standard deviation of SSS within the ensemble time period. To mimic the process
of validating satellite measurements, we also computed a mean SSS within each ensemble time period. The mean over this
interval is an approximation of the footprint mean that Aquarius would have seen in one L2 sample. For illustration, we show
the SSS record for one ensemble time period (~7 days) for one mooring in Figure 2a. The mean is also shown (red symbol in
Figure 2a), and the STV (red lines in Figure 2a). The STV in Figure 2a makes up part of the distribution for the entire record
at this location shown in Figure 2b (red line). The median value of STV for this record (green line in Figure 2b) is reported for
each mooring (Figure 3). As stated above, the validation process for satellite L2 measurements might compare them with a
single in situ measurement. To get a sense of this, we choose a random value from the ensemble to simulate a float popping
up into the satellite footprint or nearby in space or time (blue symbol in Figure 2a), and compare it with the ensemble mean.
This forms a time series of differences, summarized as a histogram in Figure 2c, over the length of the record from which we
can compute the RMS (green lines in Figure 2c). This RMS is what we will call the "RE", a single number from each mooring.
The root mean square difference (RMSD) between the "float" value and the satellite value is what is commonly reported in
validation studies (e.g. Kao et al., 2018b). In this case there is no satellite retrieval error, so the RMSD between averaged and
individual values we compute is due only to RE. The mean difference (instantaneous value - short-term mean) for each mooring
is also computed (e.g. blue line in Fig. 2c) and reported below as the bias.

## 3 Results



The median STV at each mooring (Figures 3 and 4a) mainly ranges from 0.02 to 0.15, with most between 0.02 and 0.08. In
the Pacific, the values are smallest along the equator and to the south. Larger values are found along 8°N, along 95°W in the
eastern basin at the edge of the eastern Pacific fresh pool (Alory et al., 2012), and in the western basin. The Atlantic basin has
some larger values, especially one close to the coast of Africa near the outlet of the Congo river. In the Indian basin, large
values are seen in the Bay of Bengal. This is likely due to the large input of freshwater from rivers (Akhil et al., 2020). Of all
the moorings, the median value of STV is 0.05, though the distribution of values (Figure 4a) shows a wide range.


The RE (Figures 4b and 5) is generally larger than the STV. The difference is especially notable in the southwest Indian basin.
Most values of STV lie between 0.04 and 0.14 (Figure 4b). This difference between RE and STV may be due to the presence
of outlier values in the distribution of SSS (Bingham et al., 2002; Bingham 2019). If the distribution of SSS were close to
normal, these two quantities would be about the same. This is illustrated in Figure 2c. The green bars, which represent the
RMS difference between random samples and short-term mean, are larger than one would expect from looking at the
distribution shown due to the presence of a large outlier that is not pictured in the histogram. This large value of RMS
difference, ~0.12, the green bar in Fig. 2c, is larger than the median STV for this mooring, which is about 0.03.

Though one might have expected it due to low outlier values, there is almost no bias error detected. The distribution of median
bias error is centred closely around zero, with most values less than 0.005 (Figure 4c). There is no sign of tendency for the
bias to be positive or negative. For brevity we do not show maps of bias error.

The STV computed from mooring data has been shown to be highly seasonal by Bingham (2019) for the two SPURS regions.
To understand the degree of seasonality in the variability, we have computed the average STV in each month for the entire
record at each mooring location. We display the month where STV is maximum, and the ratio of maximum STV to the
minimum STV (Figure 6). The amount of seasonality does depend strongly on location. In the Pacific, it is strong in the eastern
basin, and along the northern portion of the GTMBA, but much weaker elsewhere. The seasonality is notably weak in the
western Pacific, in contrast to the strong STV there (Fig. 3a). The Atlantic basin displays stronger seasonality than the other
two basins, especially at the western and eastern sides. The Indian basin has relatively weak seasonality in the eastern part of
the basin, but stronger in the southwest. The Bay of Bengal moorings show very little in contrast to their RE (Figure 5b).

The timing of maximum STV varies substantially from one part of the tropical ocean to another (Figure 6) and is very much
dependent on local conditions. STV is maximum in Jan-Feb in the eastern Pacific as the eastern Pacific fresh pool extends to
the west (Melnichenko et al., 2019). It is maximum in Aug-Sep under the ITCZ, mixed along the equator, and May-June in
the western Pacific. In the Atlantic, STV is maximum April-May in the eastern basin near Africa, but Aug-Sep in the western
basin. The western basin values are likely associated with the extension of the Amazon River plume into the central Atlantic
along 5-10°N (Grodsky et al., 2014). The eastern basin timing is due to the extension of the Congo River plume, which reaches





maximum extent in boreal spring (Chao et al., 2015). In the southwestern Indian ocean, the seasonality is large, but the timing is varied from Jan-Feb to April-May. In the Bay of Bengal, STV is maximum in August-September.


A set of maps of STV from the MITgcm output (Figure 7) has many similarities to ones derived from the moorings (Figure 3), however, the values are generally smaller. Larger values are found in the eastern and western tropical Pacific, the Atlantic north of the equator and the Bay of Bengal. Notably, the row of locations along 8°N in the North Pacific does not exhibit the large variability seen in the mooring data (Figure 3a). The STV in the outflow of the Congo River near the coast of Africa is

much smaller in the model than in the mooring data, possibly due to the model's use of climatological river outflow (Feng et al., 2021; Fekete et al., 2002). A similar set of maps for the RE was created for the model output, but not included here for brevity. We do include histograms of STV, RE and bias (Figure 8) for comparison with the mooring data (Figure 4). The distributions from the model are again similar to those from the moorings, but somewhat smaller. Table 1 gives median values of the distributions of STV and RE, showing larger values for the observed data. A direct comparison of STV from the

moorings vs. the model indicates that the mooring STV is larger in most cases (Figure 9), but the two have a close relationship. In only about 10% of the mooring locations is the model STV greater than the mooring STV.

STV may be mainly caused by rainfall, by internal variability in the mesoscale or submesoscale SSS field of the ocean, or by the motion of large-scale fronts (Drushka et al., 2019). It is difficult to measure these effects separately to disentangle them.

One problem with measuring the impact of rainfall on STV is that it has a strange distribution, hourly values being mostly zero even during rainy periods (e.g. Bingham et al., 2002; their Figure 11). Many of the GTMBA moorings, 88 out of 123, have precipitation measurements. As a way to measure the impact of rainfall on STV, we used those records to determine the maximum rain rate over each ensemble period. We then found the STV during periods when the maximum rain rate was greater than 1 mm / hr, and when it was less than 1 mm / hr. For almost every mooring (78 out of 88) the STV during rainy

periods was greater than for non-rainy periods. A typical example of this is presented in Figure 10. Because of the way rainfall is distributed, it does not make sense to compute correlations between maximum rain rate and STV, which can be easily seen in Figure 10. So we report the apparent connection here in this simple way, concluding that STV is indeed at least partly driven by rainfall.

A non-result that is important to report here is the lack of temporal aliasing. One might expect, within the ensemble time periods we used, that the difference between the short-term mean (e.g. red symbol in Fig. 2a) and the random samples we took (e.g. blue symbol in Fig. 2a) would increase with the difference in time between the samples and the mean times. We plotted this for each mooring and uniformly found there to be no relationship between the two. The ensemble time periods we used were too short for there to be changes in the statistics of the SSS field.




## 4 Discussion

We have computed values for STV and RE that can be factored into error budgets of satellite SSS. The values in Table 1 are typical, but there is a large range (Figures 3-5). If anything is clear from the analysis done here, it is that STV and RE depend on both time and space. In many areas studied here, especially the equatorial Pacific, STV is small and would be negligible
compared to other sources of error in L2 satellite estimates. In other areas, such as the Bay of Bengal, western North Atlantic, and eastern and western Pacific, STV is important and could play a larger or even dominant role in the error budget.

As stated in Section 2, the STV is used here as a proxy for SFV over a 100 km footprint. That is, it is the variability over a 100 km spatial scale surrounding each mooring. This use depends on the assumptions that the velocity field we used derived from
OSCAR is generally representative of that experienced by the mooring. This is needed to make the jump from our estimate of STV to that of SFV. More subtly, the region sampled is that parallel to the flow at the mooring. The scheme we have used does not sample across the flow field. Thus, we have assumed that the spatial variability across the direction of flow is similar to that along the direction of flow. Without simultaneous sampling in a spatial region surrounding the moorings (e.g. Bingham, 2019; Reverdin et al., 2015) it is impossible to know how much variation is being missed. We were able to compute SFV at
each mooring location from the MITgcm for comparison to STV from the model (Figure 11). This result shows that STV and SFV have a close relationship, and that it makes sense to use one as a proxy for the other. Figure 11 suggests that STV is about ½ of the true SFV. Thus, the values given in Table 1 as estimates of STV might be multiplied by two to get estimates of SFV at the mooring locations.

The numbers in Table 1 can be thought of as an estimate of "snapshot error" (Bingham, 2019) due to representation. This is the error in each L2 estimate captured by a SSS satellite as it passes overhead due to variability within the satellite footprint. Most estimates of SSS error are computed at L3 (e.g. Qin et al., 2020; Olmedo et al., 2020). Production of L3 values entails combining numerous individual L2 snapshots into a gridded product on a quasi-weekly or monthly basis using some form of optimal interpolation (Melnichenko et al., 2014) or bin-averaging (Vergely and Boutin, 2017). Thus, the numbers in Table 1
and Figure 4 are a worst case, errors that can be averaged out in the process of moving from L2 to L3 – assuming they are random. In a sense this is a hopeful sign. The numbers in Table 1 are much smaller than total errors associated with satellite retrieval including surface roughness, galactic reflection, etc. (Olmedo et al., 2020; Meissner et al., 2018). However, a more granular analysis, like that of Figure 5, suggests that it may not be that simple. There are times and places where REs may be significant, the eastern Pacific, Bay of Bengal and river plume regions for example. These are all regions where higher SSS
open ocean waters interact intermittently with much lower SSS coastal or river plume water. Thus, it may make sense, when computing RMS errors for satellite retrievals, to leave these areas out of the analysis, or to somehow account for the larger amounts of RE that may be present in the L2 measurements there.

There is a remarkable similarity between the STV in Figure 3 and the amplitude of the annual cycle shown by Bingham et al.
(2021 – their Figure 3). The relative sizes of the symbols are very similar in most cases. There are a few exceptions. Areas
with relatively large STV but small seasonal amplitude include the region of the South Pacific under the South Pacific
Convergence zone, some areas of the central and western South Indian Ocean, the Bay of Bengal and a couple of the moorings
in the western tropical North Atlantic. Most of these areas have small amplitude in seasonal precipitation compared to the rest
of the tropics (see Bingham et al. (2012), their Figure 11e). Thus, regions with large (small) seasonal variability are also ones
with large (small) STV. As STV appears to be somewhat driven by rainfall (Figure 9), this makes sense. Many tropical regions,
like the northern hemisphere ITCZ, with heavy rainfall are also ones with strong seasonality in rainfall as well.

The vague nature of the relationship between rainfall and STV is highlighted in Figure 10 and similar single-mooring analyses
that we do not show. The original concept for satellite SSS is that it could be used as a rain gauge (Lagerloef et al., 2008). This
may be more complicated than originally thought, at least for the short-term relationship. Work is ongoing into the use of SSS
as a rain gauge (Supply et al., 2018), i.e. a way to estimate precipitation over the ocean. Doing this with mooring precipitation
and SSS data will require a much more sophisticated approach than we have attempted here. At the very least, perhaps SSS
can be used to detect whether rain is happening or not.

**Code availability**

Code used in this publication is available from the author on request.

**Data Availability**

Data used in this publication were obtained as follows:

- OSCAR data: ESR. 2009. OSCAR third deg. Ver. 1. PO.DAAC, CA, USA. Dataset accessed [2021-01-05] at
https://doi.org/10.5067/OSCAR-03D01
- MITgcm access: https://data.nas.nasa.gov/ecco/data.php
- GTMBA data: https://www.pmel.noaa.gov/tao/drupal/disdel/



**Author contribution**

Conceptualization: FMB; Data curation: FMB & SB; Formal analysis: FMB; Funding acquisition: FMB; Investigation: FMB; Project administration: FMB; Supervision: FMB; Writing - original draft preparation: FMB; Writing - review & editing: FMB & SB

**Competing interests**

The authors declare that they have no conflict of interest.

**Acknowledgements**

Funding for this work was provided by NASA under grant #80NSSC18K1322. The authors acknowledge the GTMBA Project Office of NOAA/PMEL for use of the mooring data.

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





**Table 1:** Median values for all the moorings and mooring locations from the distributions of Figures 4 and 8.

|  | Median STV | Median RE |
|---|---|---|
| Mooring data | 0.05 | 0.09 |
| Model output | 0.03 | 0.07 |






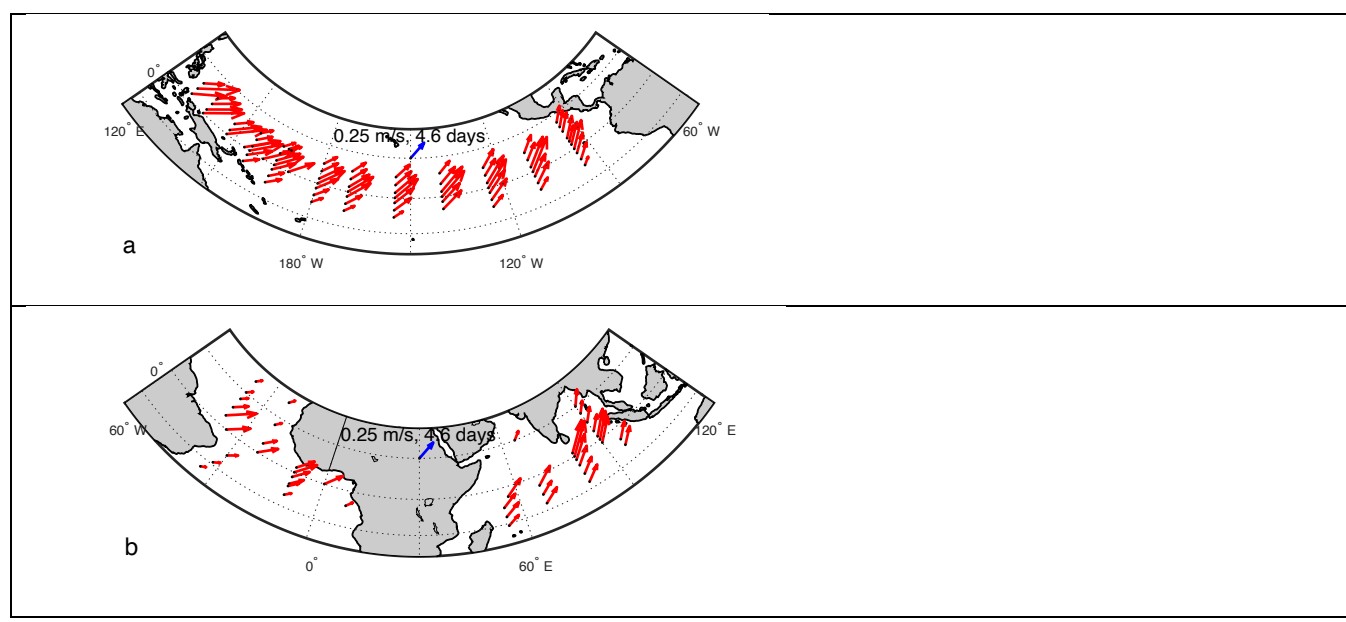

**Figure 1: Small black dots: GTMBA mooring locations. Red arrows: Mean OSCAR current speed over the 1992-2020 time period at each mooring location. Arrow lengths indicate the average speed, with a scale arrow in blue at the top center. The length of the scale arrow corresponds to 25 cm/s. At that speed, it takes 4.6 days to go 100 km. Thus, the arrow length also corresponds to the time span used in computation of short-term variability discussed in the text. A shorter arrow (slower speed) means a longer time span. The directions of the arrows have no meaning. a) Pacific basin. b) Atlantic and Indian basins.**

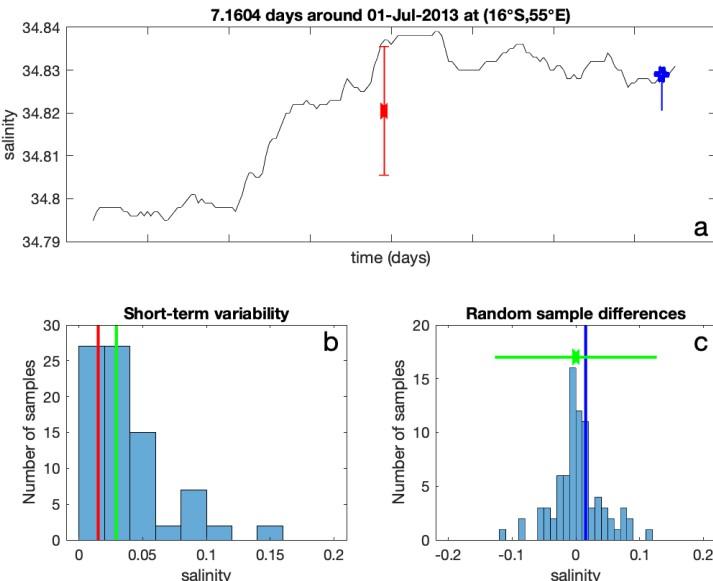

**Figure 2: An example of how the STV is computed as explained in the text. a) A ~7-day piece of SSS record from the indicated mooring. The mean value is shown by the red "X", with red bars being +- 1 standard deviation. The blue "X" is a random value picked from the record. The blue line shows the difference between it and the mean value. b) The distribution of STV from the entire record at this mooring. One value is given by the standard deviation from panel a, and is the same as the red bar in panel b. The median of the values in this distribution is indicated by the green bar. This is the single number from this mooring that is shown as**
**STV in Figure 4a. c) Distribution of differences between random samples and mean values. One value is given by the blue line in panel a, and is the same as the blue bar in panel c. The RMS of this distribution is shown by the green bars, which is the mean +- the RMS. These bars are the single value of RE that is shown for this mooring in Figure 4b.**

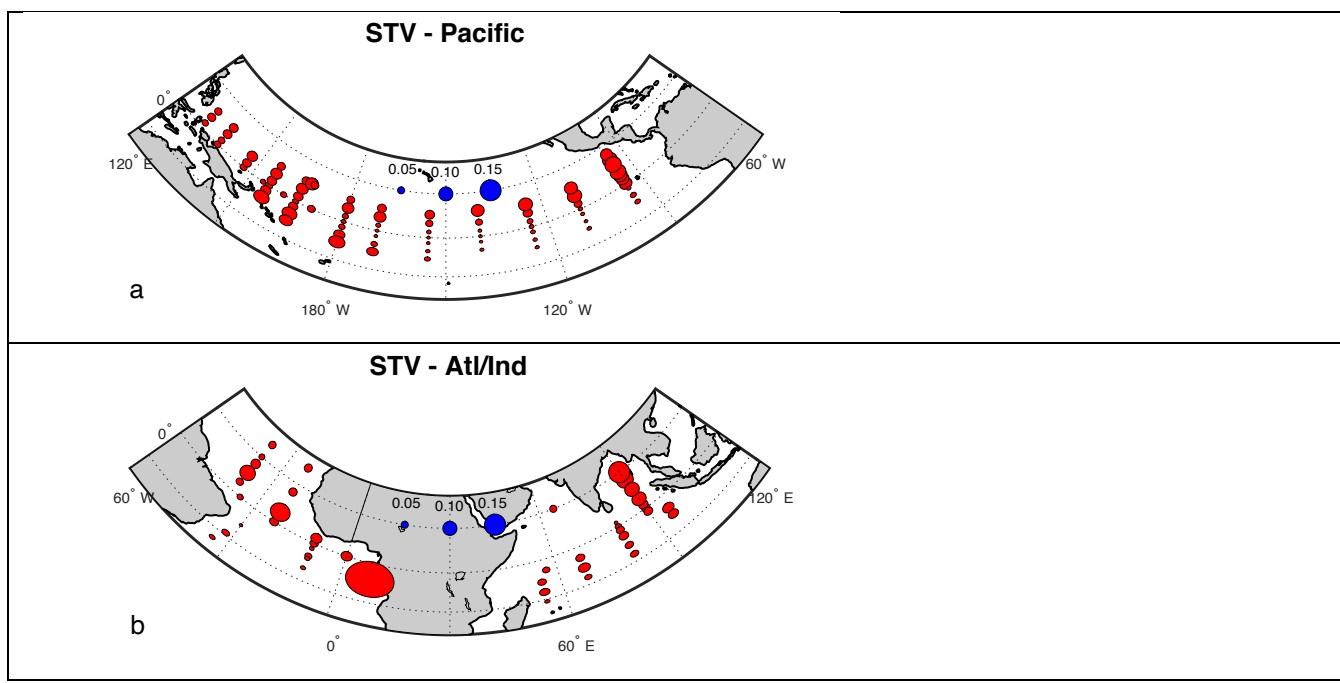


**Figure 3: Median STV for each mooring for the a) Pacific basin and b) Atlantic and Indian basins. The sizes of the circles indicate the magnitude of the STV, with scale shown as blue circles near the middle of each panel.**





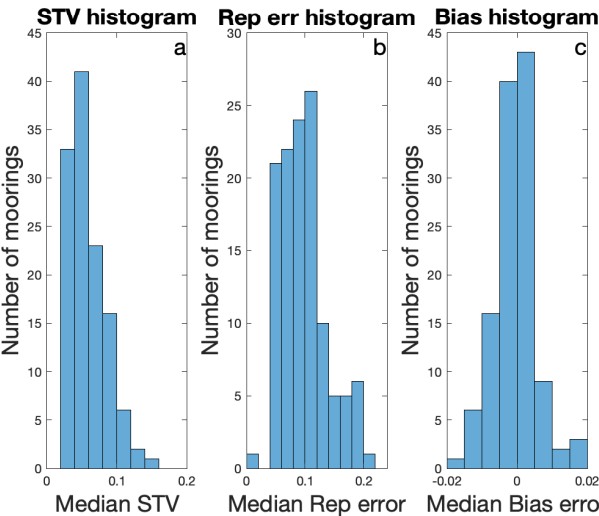


**Figure 4: a) Histogram of the median STV values shown in Figure 3. One mooring is not included in this figure, an outlier with STV~=0.3 – the large symbol near the coast of Africa in Figure 2. b) As in panel a, but for median representation error shown in Figure 5. c) As in panel a, but for bias error. Positive bias error means instantaneous value greater than short-term mean. Note panels b) and c) rely on choosing random values from each short-term ensemble. This was done a number of times with different random values with only minor differences in results. Also note, the distributions depicted here included a small number of outliers that are not shown for clarity. Note the different x-axis limits in panel c.**







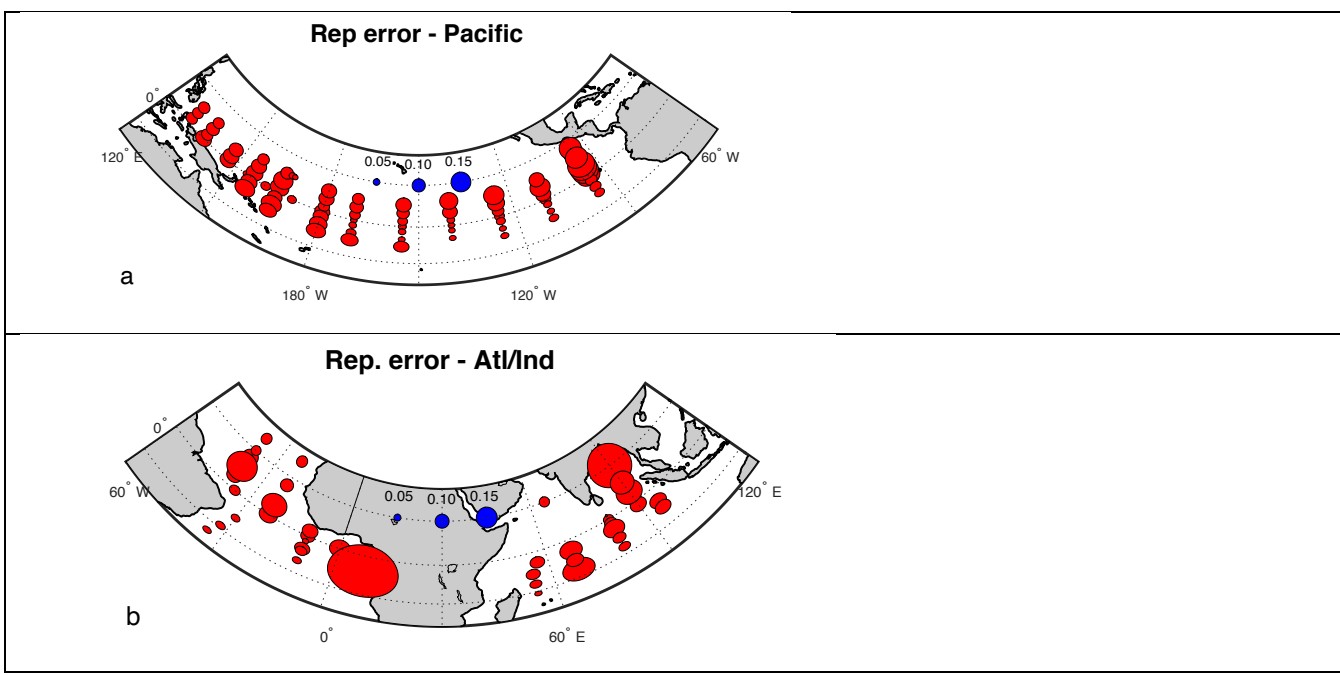

**Figure 5: Representation error at each mooring. That is, RMS difference between random samples and short-term mean values. a) Pacific basin and b) Atlantic and Indian basins. The sizes of the circles indicate the magnitude of the RE, with scale shown as blue circles near the middle of each panel.**






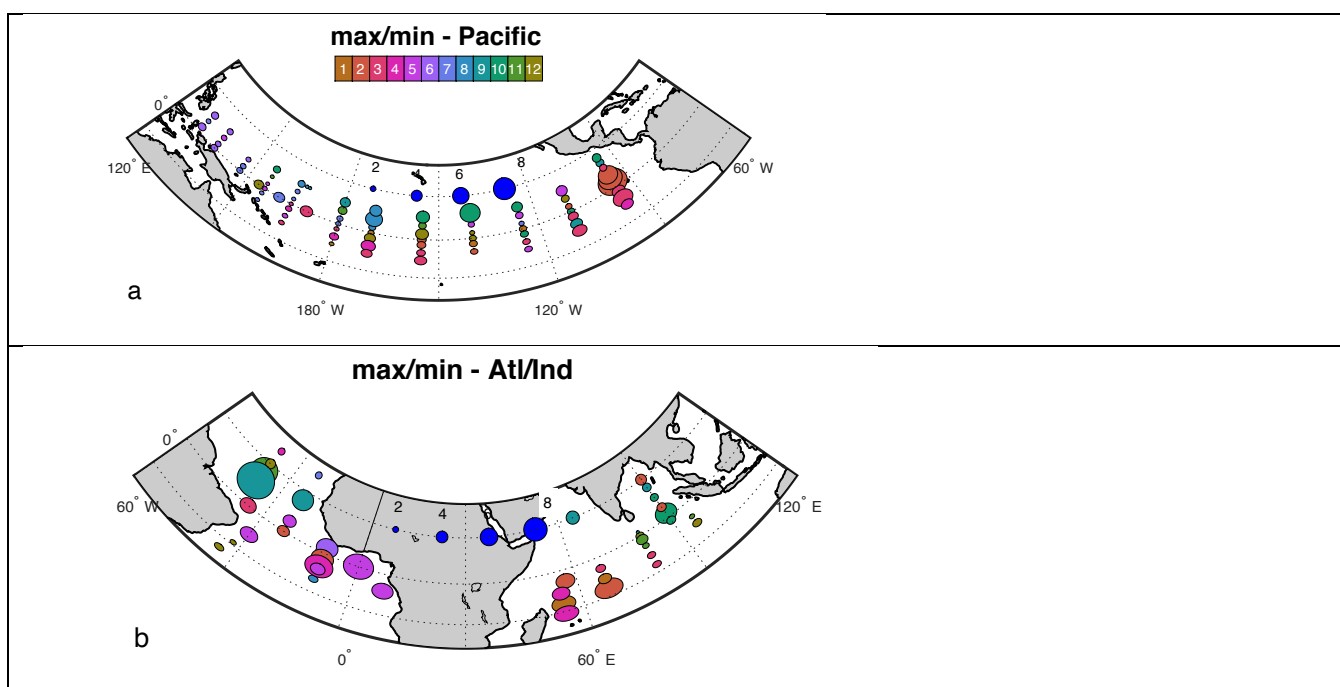

**Figure 6: Ratio of maximum value of STV to minimum value (sizes of symbols), and month of maximum STV (symbol colors with scale in panel a in months (Jan-Dec)). a) Pacific basin and b) Atlantic and Indian basins. A size scale for the STV ratio is shown as blue circles near the middle of each panel.**





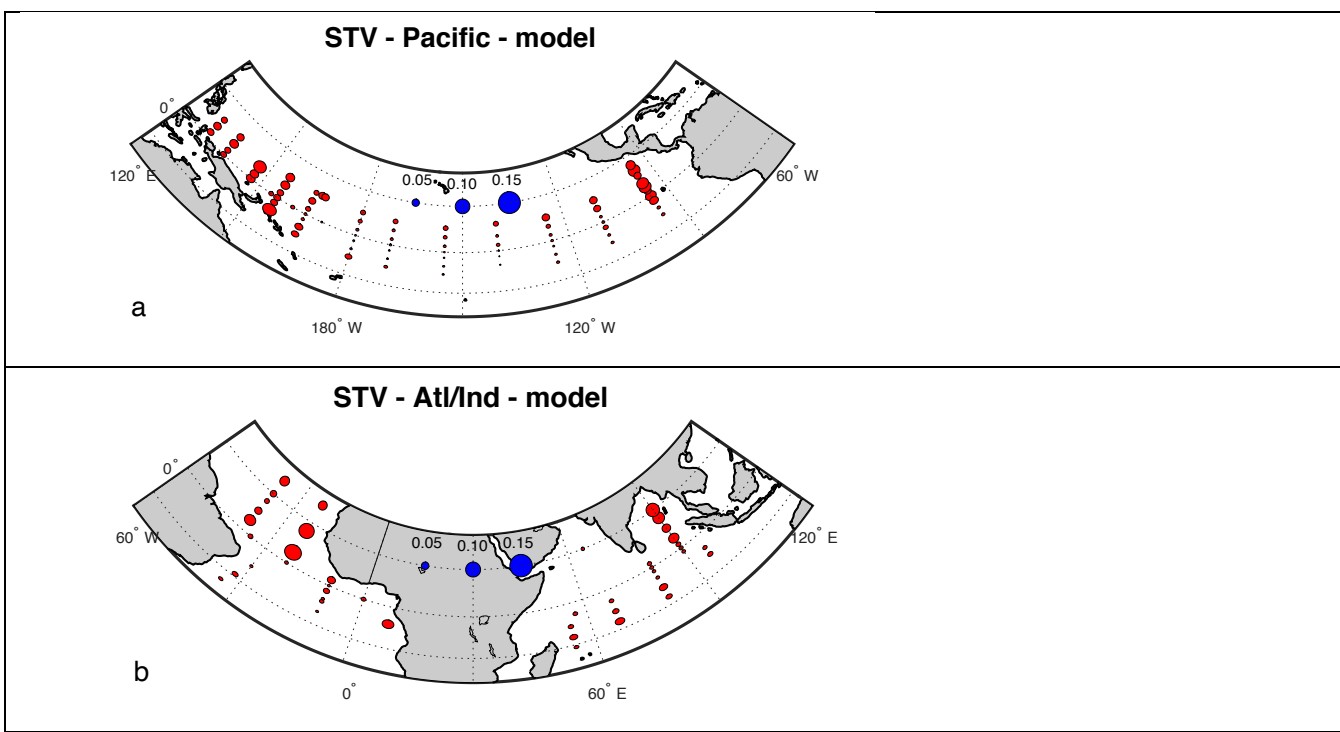

**Figure 7: As in Figure 3, but for STV computed from the MITgcm. a) Pacific basin and b) Atlantic and Indian basins.**





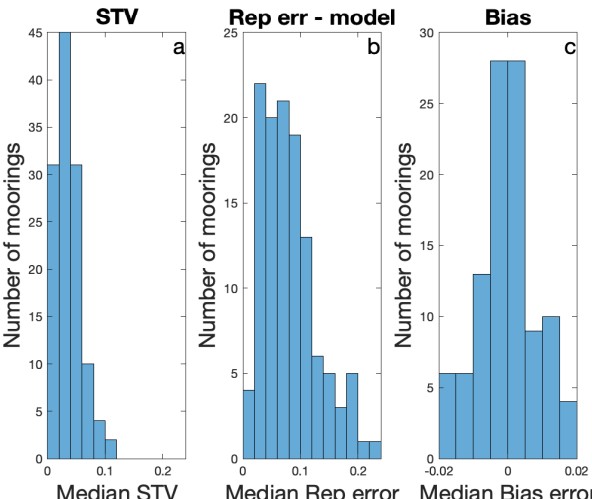

**Figure 8: As in Figure 4, but for values computed from the MITgcm.**






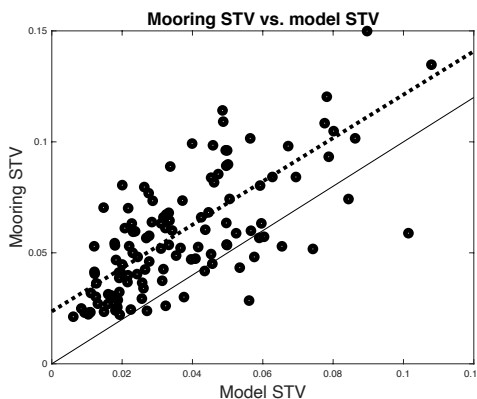

**Figure 9: Model STV vs. mooring STV. Each symbol represents the median STV for one mooring. A couple of outlier points have been omitted for clarity. The dashed line is a least-squares fit to the data. It has a slope of about 1 and an intercept of about 0.02. The light black line has a slope of 1. This plot indicates that the mooring STV is generally larger than the model STV.**







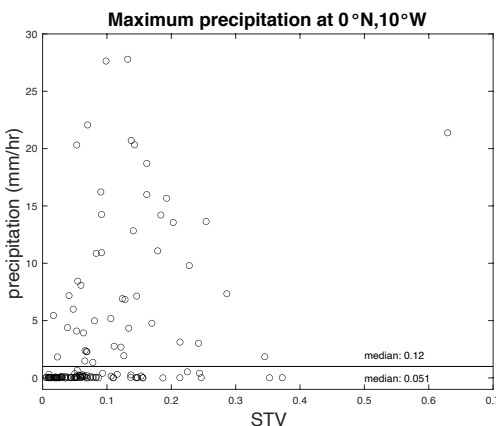

**Figure 10: An example of how STV relates to precipitation at one mooring location. Maximum hourly precipitation for each short-term ensemble for the mooring at (0°N,10°W) vs. STV for the same set of ensembles. The light line indicates rain rate of 1 mm/hr separating rainy periods from non-rainy ones. Median values for all ensembles with maximum precipitation less than 1 mm/hr**
**(below the line) and greater than or equal to 1 mm/hr (above the line) are shown at bottom right. These indicate that STV tends to be greater when there is rainfall. However, there is not a clear correlation between rainfall and STV. This pattern was consistent in most of the GTMBA moorings with precipitation records.**





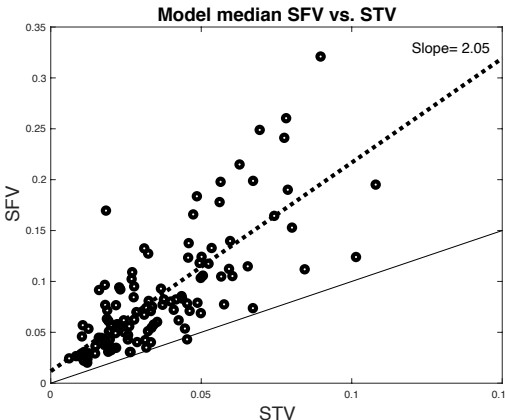


**Figure 11: Median model SFV vs. model STV at each mooring location. Each symbol represents one mooring. The light line has a slope of 1. The dashed black line is a least-squares fit to the data shown, with the slope indicated at the top right.**