# Peer review of "Sea Surface Salinity Short Term Variability in the Tropics"

_Ocean Science, 2021_

## Author Comment (AC1)

We appreciate this reviewer's kind words and thoughtful comments. Below our responses are in red.

Why choose a free-running model over a data assimilative one? In this case, it is more of an evaluation of model physics than a data-assimilative model, which might be more reflective of the actual ocean. Also, it would provide the opportunity for suggested improvements of short-term tropical variability of SSS in currently operationally used models.

This model was chosen because it is the highest resolution available, 1/48 deg. The high resolution, we hope, will make the model be reflective of the statistics of the actual ocean, which is what is most important in this case. It has been shown (Su et al., 2018) that the model does well at simulating the submesoscale variability. A data-assimilating model would certainly be closer to the "truth", but may not be as statistically robust. This was made clearer in the revised version. The suggestion about using the results to improve the current operational models is a good one, but not the purpose of this paper.

Related to my last comment, some discussion on how this work may improve parameterization of models would be interesting. Where do you see the greatest value of this work? In SSS processing? Rainfall identification?

The main purpose of this work is to understand subfootprint variability and representation error and its impact on satellite measurement of SSS. The reviewer is correct that this analysis is a good test for the model we used, or for other models. The fact that STV is lower in the model than for the moorings suggests that either the model resolution is not quite good enough to match the statistics of the real ocean, or (more likely) that the forcing fields used, especially the rainfall, are too coarse compared to the real forcing. It's clear that rainfall occurs on a scale that is smaller than the typical ocean model is exposed to. The atmosphere continually adds small scale variance to the ocean in the form of freshwater forcing. We are not experts in ocean modeling, but it would be interesting to see how the scale of the input variance affects the behavior of forced models like the one we used.

We have added text to this effect to the discussion section.

"The main purpose of this work is to understand subfootprint variability and representation error and its impact on satellite measurement of SSS. This type of analysis is a good test for the MITgcm, and could be used for other models. The fact that STV is lower in the model than for the moorings suggests that either the model resolution is not quite good enough to match the statistics of the real ocean, or (more likely) that the forcing fields used, especially the rainfall, are too coarse compared to the real forcing. It is clear that rainfall occurs on a scale that is smaller than what the typical ocean model is exposed to. The atmosphere continually adds small scale variance to the ocean in the form of freshwater forcing. It would be interesting to see how the scale of the input freshwater forcing variance affects the behavior of forced models like the one we used."

Abstract: How was 5-14 chosen? There is no other mention of it in the text.

Good question, and thanks for catching that. Changed to "2-17" days to match with section 2.2.

Lines 43-44: "SMOS (Soil Moisture and Ocean Salinity) does not have a simple footprint due to its interferometric method of sensing and wide field of view." Has anyone attempted to compute this? Is there a range of values? Is it similar at similar latitudes?

Again, this is a good question. We have been looking this for a different project. The footprint size depends on the look angle relative to nadir on the satellite, and ranges from about 35 to 63 km. There is no reference for it, and we got the information via personal communication from someone involved in the SMOS project. We put in a bit of text stating this.

"The other major SSS satellite, SMOS (Soil Moisture and Ocean Salinity) does not have a simple footprint due to its interferometric method of sensing and wide field of view, but ranges from 35-63 km depending mostly on the look angle relative to nadir (González-Haro, personal communication)."

Regarding the comparison between SFV between SPURS-1 and SPURS-2, what is the main reason for differing footprint sizes?

We presume the reviewer is referring to the paragraph on (former) lines 46-57, and the paper being referred to Bingham & Li (2020). What is different between SPURS-1 and 2 is not the footprint size, but the dependence of the SFV on footprint size. The SPURS-2 region had generally higher SFV, but less dependence of SFV on size, especially during the summer and fall (compare their Fig. 4 with their Fig. 3b). Bingham & Li discuss this observation at length, and the reader is referred there for more detail. It is possible that SFV at the SPURS-1 site is more strongly determined by rainfall than SPURS-2, even though it rains a lot less there. Or it may have to do with spatial scales of ocean internal variability being different at the two sites. In short, this is a research question that is far from being answered at the moment. We added this to the text.

"This difference may have been due to the influence of differences in rainfall and/or internal ocean variability at the two sites"

Section 2.3: I need some clarification on "weekly evaluation times." Is this via taking each mooring measurement +/- 3.5 days? Or days 1 thru 7, then 2 thru 8, then 3 thru 9, etc. for the full time period of 1992-2020 at every point? Are the initial dates of each ensemble evenly spaced throughout the year? How many ensembles are taken? Figure 2 is a clear portrayal of the analysis done for one segment, but more elaboration in

section 2.3 is needed. There's no need for methodological changes, just more explanation.

We added a figure with a month of sample record with the weekly evaluation times and ensemble time periods marked - see new Figure 2. We hope this clears up any confusion.

[Figure]

Figure 2: Sample salinity record from mooring at 1.5degS,80.5degE for the month of September 2008. The weekly evaluation times are shown by red symbols. The ensemble time period for this mooring was found to be ~3.7 days. These time periods surround the evaluation times and are indicated by the red lines, which are 3.7 days long from beginning to end. The red symbols are at the mean value of SSS for each ensemble time period.

Lines 155-160: Please add some exact numbers for "larger values"

Done.

Figure 1: If the direction of the arrows have no meaning, would it be better to color code each region with values corresponding to current magnitude (a la Fig 6)? If magnitude is the only important feature here, the arrow-length approach is difficult to clearly read. A more equilateral projection would also be easier to read and would allow for larger figures, but that is up to the authors.

Thank you for the suggestions. We decided to leave the lines as lines, but removed the arrowheads. We think this works better than sized symbols. We did change the projection as suggested to a cylindrical as opposed to a conic one. This applies to a number of figures in the paper.

---

## Author Comment (AC2)

We appreciate this reviewer's kind words and thoughtful comments. Below our responses are in red.

In the abstract the authors mention that the short-term variability computed is variability of timescale 5-14 days. But this timescale is not mentioned anywhere in the manuscript.

The other reviewer noted the same thing. We changed the abstract to be consistent with the text.

How did you manage the data gaps in the moored buoy data in your analysis? How continous is the data? Nothing is mentioned about this in the methods section.

As the reviewer points out, these mooring time series are sampled hourly, but with gaps for many of them, sometimes years in length. In our analysis, taking each ensemble time period, if there were at least 10 hourly samples within that period, we computed the STV and RE. This will be stated in a revised version.

"The GTMBA mooring time series have many gaps and missing data. The STV and RE were computed within each ensemble time period only if there were 10 or more hourly values of measured SSS."

Authors use current speed to determine the timescale of short-term variability at each mooring location. Why don't you use power/wave spectrum on the buoy timeseries (or collocated model data) to understand the timescale of short-term variability?

The current speed was used in our work to determine the approximate amount of time needed in order to sample a 100 km-sized area of ocean. (We think the reviewer is talking about computing spectra of SSS, not surface waves. The moorings did not measure the surface wave field that we are aware of.) We are in fact working on computing space/time spectra of SSS from the global model in a separate effort and hope to report on those results soon. Computing spectra of SSS from the mooring time series would make a nice future study - it's amazing that someone has not already done this! However, there are enough complications with the methodology of computing power spectra, especially given the gappy and variable

length records as the reviewer notes above, that this would add significantly to the scope of the paper and distract from the focus. Thus, we request that an effort of this type be left for the future.

Authors suggest that moorings exhibit larger short-term variability during rainy periods than non-rainy periods. Does it have seasonal variations? For example in Bay of Bengal, does this conclusion holds during both monsoon season (when there is heavy precipitation) and non monsoon seasons.

We are not sure what the reviewer is asking. There is a figure in the paper (Figure 7) showing the seeasonality of STV, when it is maximum and the ratio of the maximum to minimum value. In the BoB, the seasonality is relatively small and the phase inconsistent. The maximum STV is about 2-4X that of the minimum. One mooring has maximum STV in January and two others in September-October. We would guess that the variability of STV in the BoB is more determined by river outflow than rainfall. The reviewer may know more about this than we do. We added a short statement to this effect to the text.

"In the Bay of Bengal, STV is maximum is inconsistent, with two moorings giving maximum STV in September-October and another one in January. We suspect that STV variability in the BoB is closely related to river outflow (Akhil et al., 2014)."

Also, no description is given on how realistic is the model in capturing the surface salinity at each mooring location. A comparison (correlation & bias) with the model and buoy timeseries is lacking.

It was not stated clearly enough in the paper. The model is free-running, and does not assimilate any ocean data. Thus, there is no expectation that the model and the mooring data would be correlated or depict the same field in detail. We added a statement to the paper indicating this - below. The type of analysis we are doing, comparing the statistics in the model with those of the moorings, is in a sense a validation exercise for the model that the reviewer is looking for. Thus, we could do the additional validation

the reviewer is asking us to, but we think it would be misleading and does not reflect the model's purpose accurately.

"For this reason, it is not expected that there would be detailed agreement between model and mooring data, but the statistics of each should be similar."